



# Composition and Niche-Specific Characteristics of Microbial Consortia Colonizing Marsberg Copper Mine in the Rhenish Massif

Sania Arif[1], Heiko Nacke[2], Elias Schliekmann[1], Andreas Reimer[3], Gernot Arp[3], and Michael Hoppert[1]

[1]Department of General Microbiology, Institute of Microbiology and Genetics, George August Universität, Göttingen, 37077, Germany
[2]Department of Genomic and Applied Microbiology, Institute of Microbiology and Genetics, George August Universität, Göttingen, 37077, Germany
[3]Geoscience Centre, Department of Geobiology, Georg-August-Universität Göttingen, 37077, Germany

*Correspondence to*: Sania Arif (sarif@gwdg.de)

**Abstract.** The Kilianstollen Marsberg (Rhenish Massif, Germany) has been extensively mined for copper ores, dating from Early Medieval Period till 1945. The exposed organic-rich alum shale rocks influenced by the diverse mine drainages at an ambient temperature of 10 °C could naturally enrich biogeochemically distinct heavy metal resistant microbiota. This metagenomic study evaluates the microbially colonized subterranean rocks of the abandoned copper mine Kilianstollen to characterize the colonization patterns and biogeochemical pathways of individual microbial groups. Under the selective pressure of the heavy metal contaminated environment at illuminated sites, *Chloroflexi* (*Ktedonobacteria*) and *Cyanobacteria* (*Oxyphotobacteria*) build up whitish-greenish biofilms. In contrast, *Proteobacteria*, *Firmicutes* and *Actinobacteria* dominate rocks around the uncontaminated spring water streams. The metagenomic analysis revealed that the heavy metal resistant microbiome was evidently involved in redox cycling of transition metals (Cu, Zn, Co, Ni, Mn, Fe, Cd, Hg). No deposition of metals or minerals, though, was observed by transmission electron microscopy in *Ktedonobacteria* biofilms which may be indicative for the presence of different detoxification pathways. The underlying heavy metal resistance mechanisms, as revealed by analysis of metagenome-assembled genomes, were mainly attributed to transition metal efflux pumps, redox enzymes, volatilization of $Hg^0$, methylated intermediates of As(III) and reactive oxygen species detoxification pathways.

**Key words**

Copper mine, Ktedonobacteria, Rhenish Massif, Metagenomics, Heavy metal detoxification, Metagenome-Assembled Genome, Functional Profiling





## 1. Introduction

The historic copper mining area Marsberg is situated on the north-eastern edge of the Rhenish Schiefergebirge (Rhenish
Massif) which is composed of Variscan folded rocks of Devonian and Carboniferous age (Urban et al., 1995). The Marsberg
Upper Devonian sequence mainly consists of metamorphic clay shales, sandstones, siltstones and carbonate rocks, whilst the
Lower Carboniferous rocks contain a copper rich black shale series (Siegmund et al., 2002). Investigations of the Marsberg
copper ore deposits revealed insights in their geology, ore formation and recent re-mineralizations (Stribrny, 1987). The
copper-rich sediments formed about 380 million years ago in the Devonian on the southern edge of the Laurussia continent
(America and Europe). The Marsberg copper deposit originated from tectonic movements which caused disintegration of the
Lower Carboniferous Alum Shales and lydites and exposed the Upper Devonian rocks, resulting in fissures, faults and
breccia rich in metals (e.g., 7-16% Cu), sulphides (0.5-3.8%), carbonate carbon (0.35- 2.4 3%), and organic carbon (0.3-
2.5%) (Urban et al., 1995). In rock samples, mean contents of copper (81-1277 ppm) are found to be higher than those of
other metals (Pb 36-417, Zn 78-660, Co 34-63, Ni 79-450, V 49-160, and Cr 45-193 ppm) (Urban et al., 1995). The Upper
Devonian to Lower Carboniferous rocks are completely exposed in Marsberg Kilianstollen copper mine, offering conditions
for formation of diverse secondary minerals and mine drainages. The most important sulphide ore minerals present in fault
and fault-related breccia zones are described as chalcopyrite, neodigenite, chalcocite, bornite, and covellite (Stribrny, 1987).
These minerals in the sediments all shape the prevailing biogeochemical conditions in the Marsberg mine waters. The
biological and atmospheric oxidation of sulphides from pyrite ($FeS_2$) or chalcopyrite ($CuFeS_2$) could mobilize transition
metals (Amin et al., 2018). The reduced transition metals (Cu, Fe, Mn) from fault-bound breccias and black alum shales are
oxidized, resulting in copper-rich (acidic) sulphidic mine waters with high concentrations of copper and iron, but also other
dissolved ions (Fig. 1, green color). In addition, the fresh groundwater (flowing in NE direction) is enriched with calcium
carbonate from the Upper Permian Zechstein limestone, while sulphate is derived from gypsum and anhydrite of the same
formation group (Fig. 1, blue color). The calcium and sulphate levels in these mine waters can be higher than in fresh water,
up to 2/3 of the values for sea water. These naturally flowing water streams of the Kilianstollen mine ranging from fresh
water to heavy metal enriched acidic leachate offers unique subsurface cold heavy metal enriched habitat to study the
colonized microbial communities under influence of various mine waters.



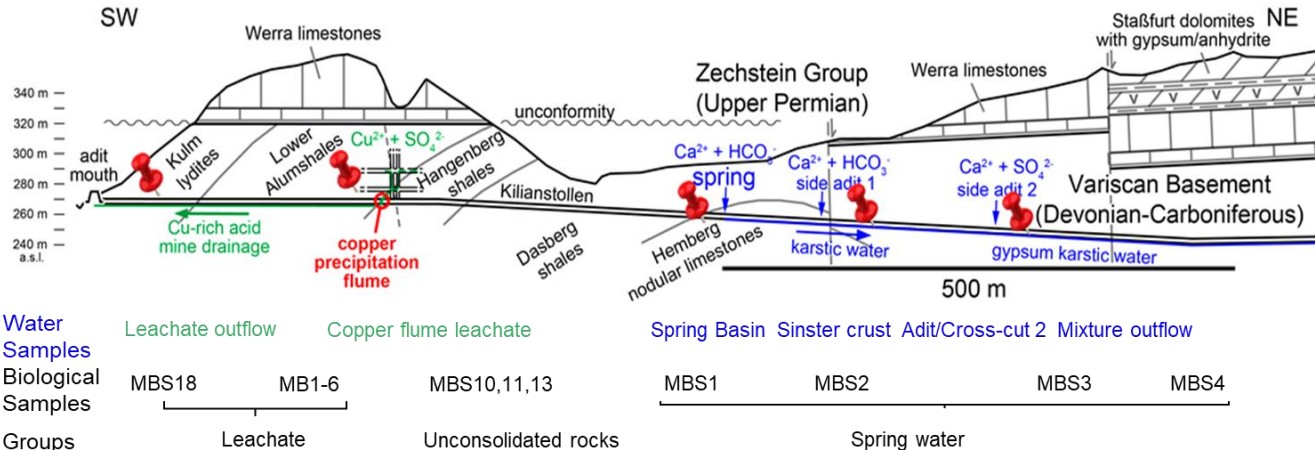

Figure 1. Simplified geological section of the Kilianstollen (Marsberg) and formation of different ground waters. Folded
Devonian-Carboniferous lydites, shales, and nodular limestones of the Variscan Basement are unconformably overlain by
Upper Permian carbonates and evaporites of the Zechstein Group. The reduced transition metals (Cu, Fe, Mn) from fault-
bound breccias and black shales are oxidized, resulting in copper-rich (acidic) sulphidic mine waters (Emmerich, 1987),
draining towards the adit entrance in SW direction. Some of the mine waters (flowing in NE direction) are enriched in
calcium, carbonate, and sulphate ions, depending on the rocks (limestone and gypsum) in contact. The geological section is
based on (Oskar and William, 1936; Farrenschon et al., 2008; Stribrny, 1987; Stribrny and Urban, 2000), and an unpublished
mining map. The red pins mark the location of the sampling sites along the mine drainage system for both water and
biological samples collections. The biological samples were further grouped into three groups based on their origin and
nearby waters.

An ambient temperature of 10 °C, a relative humidity of 98% and an appropriate abundance of organics (2-10%) in the alum
shale, these organic-rich copper shales also provide microscale spaces for microbial colonization and aromatic compounds
catabolism, perhaps because of their high content of kerogen, providing partially complex organic biodegradable compounds
such as long-chain and polycyclic aromatic hydrocarbons, esters, organic acids, thiophenes and metalloporphyrins (Dziewit
et al., 2015). The availability of the soluble sulphate ($SO_4^{2-}$) and transition metals ions from the nearby sulphuric waters
sources (Silver and Walderhaug, 1992) are important in shaping an epilithic but also heavy metal and/or acid-tolerant
bacterial community. Nevertheless, the emissions of the operational Marsberg mine railway diesel engine also provide
another resource of organic compounds and the regular visits of tourists, artificial ventilation and illumination are some man-
made impacts on the native microbial consortium. In this study, prokaryotic communities associated with the rocks around
the Marsberg Kilianstollen mine waters were metagenomically evaluated to observe whether the mine waters enriched in
transition metals may be toxic to microbial inhabitants or, conversely, support unique forms of metal respiration and enrich
resistant microbial consortia under oligotrophic conditions. To elucidate further key processes involved in their resistance





against high transition metal concentration and metabolism of the aromatic compounds, the metagenomically assembled genomes (MAGs) from a biofilm (MB1) nearby the copper-rich (acidic) sulphidic mine waters were assembled and analysed for the genetic targets related to toxic Hg(I) and As compounds reduction, Cu(I) oxidation, heavy metal ions extrusion,

dehalogenation, and hydrocarbon compounds catabolism. Understanding the selective pressure exerted by heavy metals on microbes and corresponding microbial resistance mechanisms could unveil their biogeochemical consequences and applications.

## 2. Materials and Methods

### 2.1 Sampling site

The rock samples colonized with soft biomass/biofilm around mine drainages of the Kilianstollen, Marsberg, Germany (51.453502°N, 8.861703°E) were collected under sterile precautions: Hammer and chisel were disinfected with 70% ethanol prior to use, gloves were worn during sampling in order to reduce risks of contamination (Fig. 1, S1). The samples were taken from two locations in the alum shale region where mining activity was particularly high. For statistical analysis, the

samples (MB1-6 and MBS18), (MBS10,11,13) and (MBS1-4) were divided into leachate, unconsolidated rocks and spring water groups based on the nearby drainage water bodies and some of the representative samples are shown in Fig. 2. A detailed  description of the biological features and origin of the collected samples has been published previously (Arif et al., 2021b). The biofilm and water samples were taken along the natural flowing water streams in the Kilianstollen mine, for instance, the biofilms (MBS1-4 grouped as spring water biofilms) were collected around the spring freshwater stream which

was gradually enriched in calcium, carbonate, and sulphate ions with the flow streams from the crosscut that branches off from the rear course of the mine (Fig. 1). The biological samples under the influence of the copper-rich acidic leachate stream were taken from the immediate vicinity of a copper precipitation flume 'Zement-Kupferplatte' (MB1-6 biofilms) and directly from the entrance of mine outflow stream (MBS18), were grouped as leachate samples (Fig. 1, 2). The copper precipitation flume is an iron plate which is continuously flooded by the copper-rich acidic leachate stream and the copper is

being deposited electrochemically.  The other samples from a wooden plank (MBS10) next to mine unconsolidated rocks (MBS11, 13) (Fig. 2) were grouped as unconsolidated samples. Samples collected for microscopic analysis were refrigerated until preparation. For extraction of metagenomic DNA, freshly collected samples were stored at -20 °C till further use. The samples (0.4 g each in triplicates) for DNA extraction were obtained by scratching the biofilms of the rock piece and wooden plank with a sterile scalpel.





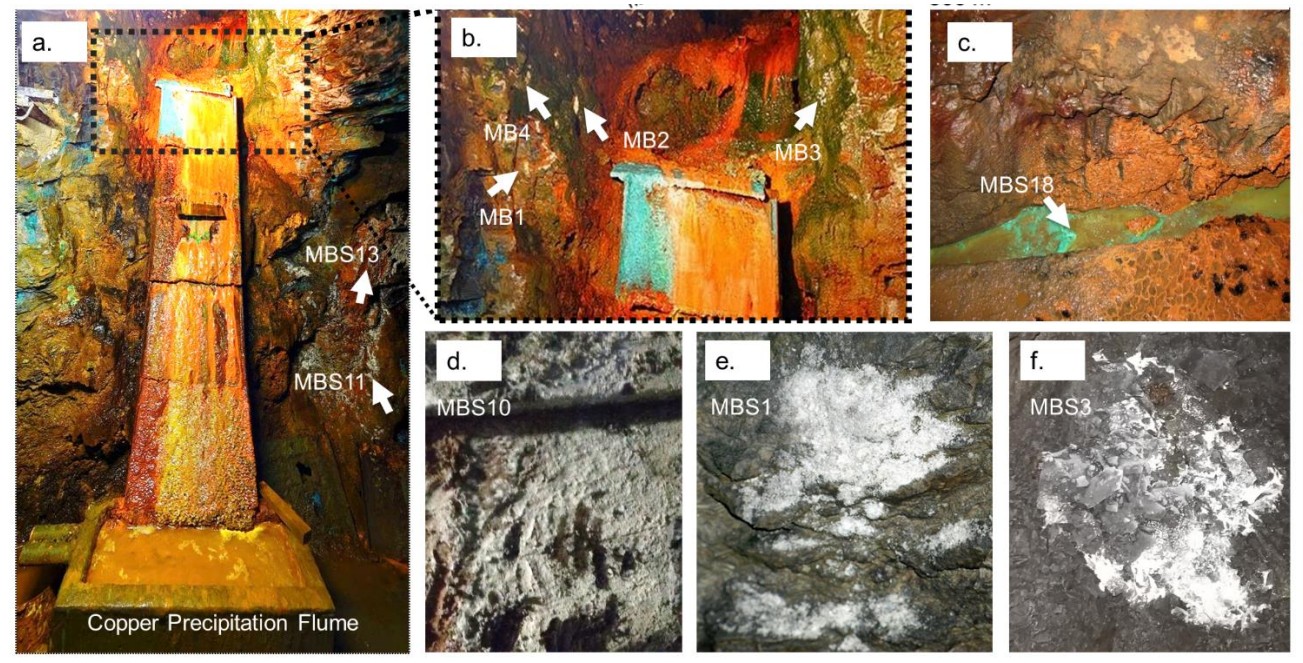

Figure 2. Kilianstollen biological samples. Samples were taken from sites in the vicinity of a copper precipitation flume (a and b) that is drained actively with the leachate water, from a wooden plank (d) located just right to the copper flume, directly from the outflow stream (c) near the opening and from rocky mine walls (e and f) exposed to the underlying groundwater and pit water stream at the rear end crosscut of the Kilianstollen.

## 2.2 Hydro-chemical analysis

Mine drainage water around sampling sites was analysed on site via a WTW Tetracon 925 conductivity probe equipped with a WTW Multi 3430 device, a WTW Sentix 940 electrode for pH and temperature, and a Schott PT61 redox electrode (Xylem, Rye Brook, NY, USA); calibration was performed with pH buffers 12.000, 10.010, and 7.010 (HI6012, HI6010 and HI6007, Hanna Instruments, RI, USA).

To determine the anions, cations, total organic carbon (TOC), total alkalinity (TA) and dissolved inorganic carbon (DIC), water samples were collected without headspace in Schott-Duran glass bottles (Schott, Mainz, Germany) and polyethylene (PE) bottles. Samples considered for cation analysis were filtered in separate 50 mL aliquots through 0.7 µm pore membrane filters and acidified with 100 µl $HNO_3$ (Suprapur; Merck, Darmstadt, Germany). For determination of total sulphide ($\Sigma H_2S$), aliquots were fixed with Zn-acetate. Total alkalinity (TA) was determined by acid-base titration within 2 hrs after sampling using a hand-held titration device and 1.6 N $H_2SO_4$ cartridges as titrant (Hach Lange GmbH, Düsseldorf, Germany). All other samples were processed within 24 hrs after sampling.

Main cations ($Li^+$, $Na^+$, $K^+$, and $Mg^{2+}$, $Ca^{2+}$, $Sr^{2+}$) and anions ($F^-$, $Cl^-$, $Br^-$, $NO_3^-$ and $SO_4^{2-}$) were analysed by ion chromatography with non-suppressed and suppressed conductivity detection (Metrohm 820 IC, Metrohm 883 Basic IC;



Metrohm, Herisau), respectively. The concentrations of $NH_4^+$, $NO_3^-$, $PO_4^{3-}$, $\Sigma H_2S$ and dissolved silica were determined
photometrically, using a SI Analytics Uviline 9400 spectrophotometer. Dissolved inorganic carbon was determined with a
TOC-L CPH analyser (Shimadzu, Kyoto, Japan). The PHREEQC software (version 3.5.0, 2019; (Parkhurst and Appelo,
2013)), with the phreeqc.dat and wateqf4.dat databases was used for calculation of ion activities, $pCO_2$ (partial pressure of
$CO_2$) of samples and mineral saturation states. Saturation is given as $SI = \log (IAP/KSo)$, where IAP denotes ion activity
product and KSo solubility product of the mineral phase.

## 2.3 DNA extraction, 16S rRNA gene amplification and amplicon sequencing

The microbial genomic DNA (gDNA) from 0.4 g scratched samples were extracted by using the DNeasy PowerSoil kit
(Qiagen, Venlo, the Netherlands) as per manufacturer's instructions. In brief, the total gDNA released from cell lysis was
treated for inhibitors removal and protein precipitation, then captured on and subsequently eluted from a silica membrane of
a spin column. Blanks were also processed in addition to each sample to estimate DNA contamination. Following elution,
the extracted gDNA was visually observed with 0.8% agarose gel electrophoresis using TAE buffer, pH (8.3) (Sambrook and
Russell, 2001) and photometrically quantified using a Nanodrop ND-1000 spectrophotometer (PeqLab, Erlangen, Germany).
No gDNA contamination was measured in the blanks following the DNA extraction.

For Illumina MiSeq sequencing, V3-V4 hypervariable regions of 16S rRNA genes were amplified via polymerase chain
reaction (PCR) and tagged to 5′ overhang adapter sequences (underlined) with the aid of MiSeq 16S amplicon PCR forward
primer 341F 5′-TCGTCGGCAGCGTCAGATGTGTATAAGAGACAGCCTACGGGNGGCWGCAG-3′ and reverse primer
805R 5′-GTCTCGTGGGCTCGGAGATGTGTATAAGAGACAGGA CTACHVGGGTATCTAATCC-3′) (Klindworth et
al., 2012). PCR reaction mixture (Amin et al., 2018) was modified to obtain a final volume of 50 μl in double-distilled
nuclease-free water by mixing 5 × Phusion GC Buffer (10 μl), 10 μM forward and reverse primer (1.0 μl each), 10 mM
dNTPs (1.0 μl), 5% DMSO (v/v, 2.5 μl), 50 mM $MgCl_2$ (0.15 μl), 0.5 μl of 2 U/μl Phusion HF DNA polymerase (Thermo-
145 Fisher Scientific, Waltham, MA, USA) and 25 ng template DNA (2.0 μl). The PCR profile comprised preheating at 94 °C
for 3 min followed by 25 cycles of heating at 94 °C for 45 s, annealing at 60 °C for 45 s and extension at 72 °C for 30 s. The
reaction ended with a final elongation step at 72 °C for 5 min. After PCR amplification, the amplicons were visually
assessed with gel electrophoresis using 1.3% (w/v) agarose in 1× TAE buffer (Thermo-Fisher Scientific), pH 8.3 (Sambrook
and Russell, 2001), and photometrically quantified in a Nanodrop ND-1000 spectrophotometer (PeqLab). The subsequent
purification was performed with the GeneRead Size Selection Kit (Qiagen) to remove primers and PCR reagents. After
indexing of these PCR amplicons using the Nextera XT DNA library prep kit (Illumina, San Diego, CA, USA), paired-end
sequencing was performed with an Illumina MiSeq sequencer in collaboration with the Göttingen Genomics Laboratory.

### 2.3 Amplicon sequencing data processing

The Illumina amplicon sequencing data was processed online by employing the automated pipeline for metagenomic
analysis MetaAmp (http://ebg.ucalgary.ca/metaamp/) (Dong et al., 2017). Using USEARCH software package, the





demultiplexed Fastq format sequence files were assembled as paired-end reads (Edgar, 2010). The misaligned and mismatched reads and paired end reads shorter than 350 bp length were discarded. Next, the primers were trimmed based on the Mothur software package (Schloss et al., 2009), and the reads without primers or with mismatched primer regions were removed. To minimize sequencing errors, the low-quality reads were scrapped using USEARCH. Dereplication, removal of singletons and chimeras, and clustering of pooled high-quality reads into operational taxonomic units (OTUs) on the basis of 97% identity was done by UPARSE software (Edgar, 2013). The taxonomic status of the OTUs was assigned via Mothur by using the SILVA v138 as a reference (Glöckner, 2019). The taxonomic profile obtained from MetaAmp was further fed to the Microbiome online R-based server to plot respective graphs (Arndt et al., 2012). Based on the Bray-Curtis index, in principal coordinates analysis (PCoA) the data representing similarities of complex microbial communities were plotted into 2D and 3D graphs. Amplicon sequencing data has been published to the Sequence Read Archive (SRA) as SRR12876542-SRR12876555 under the Project accession number PRJNA670497 as announced previously (Arif et al., 2021b).

16S rRNA gene sequences related to already published *Ktedonobacteria* and *Actinobacteria* strains were included for phylogenetic analysis. In a first step, sequences belonging to *Ktedonobacteria* or *Actinobacteria*-related OTUs were aligned with the already published sequences through MUSCLE, implemented in MEGA-X software, by using default settings (Stecher et al., 2020). Next, following the Kimura 2-parameter model, phylogenetic analyses and molecular evolutionary distances were calculated. The phylogenetic trees were constructed using the maximum likelihood algorithm and 1,000 bootstrap samplings to test tree topology.

**2.4 Microscopy**

The topographical features of the sampled biofilms (MB1-6) from the leachate group were observed using a Motic SMZ-171 stereo microscope (Motic GmbH, Germany) equipped with a Canon A650 camera. For transmission electron microscopy (TEM), the biofilm MB2 specimens were washed with phosphate buffered saline (50 mM, pH 7.0–7.5), followed by fixation in 2% (v/v) glutaraldehyde solution and incubation at 0 °C for 90 min. Subsequently, samples were dehydrated in a series of 15%, 30%, 50%, 70%, 95%, and 100% (v/v) aqueous ethanol solutions each for at least 30 min. After embedding samples with 66.6% LR White resin (London Resin CO Ltd., UK) in ethanol at 25 °C for 2 h and overnight incubation in 100% resin at 4 °C, the samples were polymerized for 12 h at 55 °C. The 80-100 nm ultrathin sections were cut with diamond knives (DDK, Wilmington, DE, USA) in Reichert Ultracut E ultramicrotome (Leica Biosystems, Wetzlar, Germany). The sections stabilized by formvar-coated 300 mesh copper grids (Plano GmbH, Wetzlar, Germany) were stained with Uranyl Acetate Replacement Stain (Electron Microscopy Sciences, Hatfield, PA, USA) for 20 min. Images were captured with a Gatan Orius 4 K camera attached to a Jeol 1011 electron microscope (Jeol GmbH, Munich, Germany) and processed with the 314 Gatan Digital Micrograph software (Gatan Inc., Pleasanton, USA) and Adobe CS2 Photoshop (Adobe Systems Inc., San José, Cal., USA).





**2.5 Metabolic profiling based on Metagenome-Assembled Genomes (MAGs)**

Extracted DNA from the biofilm sample MB1 abundant in Chloroflexi, Cyanobacteria and Actinobacteria as a representative of leachate group was submitted to the Göttingen Genomics Laboratory for shotgun metagenomic sequencing. The gDNA
extraction followed by quality and quantity assessment was performed as described above. Illumina paired-end sequencing libraries were prepared using the Nextera DNA sample preparation kit and subsequently sequenced on a MiSeq system with the reagent kit v3 with 600 cycles (Illumina). For pre-processing of sequencing data, quality control, per-read quality pruning, read filtering, adapter trimming, and base correction fastp v.0.19.4  (Chen et al., 2018) was used. The assembly of short read metagenomic data into metagenomic scaffolds was carried out by the metagenome assembler metaSPAdes
v.3.14.0 (Nurk et al., 2017). Subsequently, bins were determined using MaxBin v.2.2.7 (Wu et al., 2015). CheckM v.1.1.2 was used to evaluate the MAGs quality by providing robust estimates of genome completeness and contamination (Parks et al., 2015). Each high-quality MAG was then annotated using PROKKA v1.14.5 (Seemann, 2014). Genome wide orthologous clusters across multiple species were determined with a web server: OrthoVenn v2 (Xu et al., 2019), which assigned the protein sequence data to a high-level summary of functional categories such as biological process, molecular function, and
cellular component with GOSlim annotation and UniProt search. Finally, the PROKKA output was analysed by using the PathoLogic (Karpe et al., 2011) component of the Pathway Tools software v.23.5 (Karp et al., 2015) and the MetaCyc database v.23.5 (Caspi et al., 2019). The Metagenome-Assembled Genomes (MAGs) were classified taxonomically using GTDB-Tk v.1.0.2 and the Genome Taxonomy Database (GTDB) (release 89) (Chaumeil et al., 2020; Parks et al., 2019). The raw sequencing and assembly data have been already published in the SRA (SRR12886061) and Genbank
(JADEYI000000000 and JADMIG000000000-JADMN000000000) under the Project accession number PRJNA670497 (Arif et al., 2021a). The pathways maps of these MAGs showing their metabolic potentials and annotations can also be assessed at the Göttingen Research Online Database https://doi.org/10.25625/W9PWCX.

**3. Results**

**3.1 Physiochemical parameters of Kilianstollen mine waters**

The physiochemical parameters pH, electric potential and major ions concentrations were determined for the spring water samples (spring basin; source of fresh ground water), and after the intermixing of a stream rich in calcium/hydrogen carbonate (side adit 1), where a solid sinter crust was formed in the stream bed. Two other records of parameters were taken in a stream rich in calcium/sulphate ions (side adit 2, "gypsum karstic water"), before and after mixing with the karstic water of the mainstream. Further, two samples were taken from a copper flume leachate and a copper sulphate leachate, outflow
towards the adit mouth (Fig. 1). The concentration of dissolved ions indicated higher contents of Ca, Na, Cl, SO4, Fe, Cu, Zn, and Mn in the leachate samples (Fig. 3). The highest concentration of the transition metals Ni, Co, Fe, Cu, Zn, and Mn (0.62, 0.46, 35, 85, 2, and 4 mgL-1), was observed in the acidic copper flume leachate (pH 4.8) along with $SO_4^-$, $Ca^+$, $Na^+$, $Cl^-$ and $NO_3^-$ (1952, 403, 415, 499, and 188 mgL-1) (Fig. 3a). Due to the metal precipitation, the concentrations of these





transition metals towards the adit leachate outflow stream dropped to 0.2, 0.1, 2.9, 0.005, 0.8 and 1.3 mgL-1, respectively,

and pH raised to 7.26. The heavy metal content in spring water was considerably lower, i.e., in the range of 0.23 to 5.9 µgL-1. The efflux of the transition metal ions from the adit crosscut 2 drainage raised the heavy metal concentration of the spring water in the range of 3.8 to 262 µgL-1, particularly for Zn, Mn, Cu, and Ni (262, 76, 44 and 18 µgL⁻). To understand the copper toxicity and homeostasis with respect to microbial consortia, biofilms growing at the rocky mine walls were investigated nearby these water bodies.

Figure 3 (a): Bar chart titled "a." with y-axis "mgL⁻¹" from 0 to 2000. Legend: Ca, Mg, Na, K, Sr, Cl, SO4, NO3, F, Br. X-axis categories: Spring basin, Sinter crust, Cross-cut 2, Mixture outflow, Leachate outflow, Copper flume leachate. Inset box labeled "pH" with values 7.35, 8.22, 8.08, 8.166, 7.26, 4.87.

Figure 3 (b): Bar chart titled "b." with y-axis "µgL⁻¹" from 0 to 2000 (left) and 0 to 90000 (right). Legend: Ba, Li, Co, Ni, Mo, Fe, Cu, Zn, Mn. X-axis categories: Spring basin, Sinter crust, Cross-cut 2, Mixture outflow, Leachate outflow, Copper flume leachate.


Figure 3. Heavy metal content in Marsberg drainage mine waters. The concentration of major ions related to alkali and halogen groups (a) and heavy metals (b) have been compared. The insert shows the pH of each mine water, in the same order as depicted for major ions. The copper flume heavy metal measurements were separately plotted as the copper and iron ion concentrations were exceptionally high. The adit crosscut 2 mine drainage mixed with the sinter crust stream to form the

mixture outflow stream.



## 3.2 Distribution of bacterial taxa in Marsberg Kilian copper mine samples

The distribution of the predominant taxa at the collection sites varied drastically with the quality and type of nearby water sources (Fig. 4a). The most obvious sign of microbial growth, visible to the naked eye, were sub-aerial, whitish biofilms (MBS1-4) growing on the rocks, nearby the spring water stream. They were dominated by *Proteobacteria* (38%) and

*Actinobacteria* (21%). The relative abundance of *Proteobacteria* declined, whereas the relative abundance of *Actinobacteria* increased gradually as the sampling site moves from spring water (MBS1) to karstic water containing heavy metal discharge influx (MBS3). In the leachate samples group, *Chloroflexi* (30%), *Cyanobacteria* (23%) and *Actinobacteria* (19%) were abundant in the greenish-whitish biofilms (MB1-6) collected either in close vicinity of the copper flume leachate or directly from the heavy metal leachates streams (outflow water stream sample MBS18). Since these sites were more intensively

illuminated with light bulbs than other sampling sites, this could have facilitated the growth of *Cyanobacteria*. The biofilms collected next to the copper flume from wooden plank (MBS10) and moist unconsolidated rocks (MBS11,13) were enriched mainly in *Actinobacteria* (41%) and *Acidobacteria* (20%). When the three sample groups were compared in terms of abundant taxa, *Cyanobacteria* and *Proteobacteria* were significantly abundant in the leachate and spring water stream group samples, respectively ($p<0.05$ ANOVA), while *Actinobacteria* and *Acidobacteria* seemed to be ubiquitous.

At the class taxonomic rank, 5657 OTUs were identified across all samples comprising *Actinobacteria* (25%), *Ktedonobacteria* (13%), *Oxyphotobacteria* (12%), *Acidobacteria* (9%) *Gammaproteobacteria* (8%), *Deltaproteobacteria* (5%), *Bacteroides* (5%) and *Deltaproteobacteria* (3%). MBS1-4 whitish biofilms growing nearby a spring water stream ere dominated by *Deltaproteobacteria* and *Gammaproteobacteria* (48%), *Bacilli* (29%), and *Actinobacteria* (51% and 25%). *Ktedonobacteria*, being the most abundant class (26%) in the leachate samples group (MB1-6 and MBS18), constituted 85%

of *Chloroflexi*. *Oxyphotobacteria* (a class within the phylum *Cyanobacteria*) and *Actinobacteria* also contributed 23% and 16% to the leachate group. Since the corresponding biofilms colonize rock surfaces in direct contact to the acidic as well as sulphidic and transition metal (Fe, Cu, Zn, and Mn) rich mine drainage water (Fig. 1 and 2), it is assumed that the low pH and the high heavy metal concentration of mine water contributed to the *Ktedonobacteria*, *Oxyphotobacteria*, and *Actinobacteria* natural enrichment. Statistical analysis revealed that the classes *Ktedonobacteria* and *Oxyphotobacteria* were

significantly abundant in the leachate group, while the *Bacilli* in the spring water stream group and *Actinobacteria*, *Acidobacteria Fimbriimonadia*, and *Gemmatimonadetes* classes in the unconsolidated rocks group were significantly abundant as compared to other groups classes ($p<0.05$ ANOVA).











Figure 4. Relative abundances of Marsberg Kilianstollen bacterial taxa. At phylum (a) and class (b) level, the taxonomy and relative abundance of the OTUs depicts the bacterial community composition and colonization at Kilianstollen sampling sites. The samples are shown in three groups based on their origin (unconsolidated rocks, leachate and spring water). The classes showing less than 2 % relative abundance are not mentioned. Besides bacterial taxa, the selected primers also led to the detection of *Crenarchaeota*, *Euryarchaeota*, and *Thaumarchaeota* in low abundance.

### 3.3 Alpha and beta diversity

The alpha diversity index Chao1 indicated that unique OTUs (richness) were abundant in the MBS1-4 spring water stream samples, followed by the MBS10,11,13 unconsolidated rocks and MB1-6, MBS18 leachate group samples (p<0.0003 ANOVA) (Fig. 5). The Shannon diversity index showed the same pattern when the sampling groups were statistically compared ($p$ =0.025 ANOVA, Table S1). Alpha diversity is high in the MBS1-4 samples, possibly due to moisture and neutral conditions from the adjacent spring water stream. With respect to the leachate samples group, low diversity indexes were observed as only a few adapted microbes could colonize, indicating an enrichment effect due to extreme environmental conditions (Fig. 5a).

The principal coordinates analysis (PCoA) showed that the microbial communities of spring water, leachate, and unconsolidated rocks samples were distinct to each other (Fig. 5c, d). According to the Unifrac weighted algorithm (Lozupone et al., 2011) and analysis of molecular variance (AMOVA) nonparametric method (Table S2) (Mengoni and Bazzicalupo, 2002), the spring water and leachate samples were phylogenetically distinct to each other ($p$ =0.034), conversely, the unconsolidated rocks microbiome was similar to both spring water and leachate groups ($p$ >0.097). Conclusively, the unconsolidated rocks group is the intermediate between the other groups in terms of diversity and the diverse environmental conditions led to enrichment of different microorganisms. The hypotheses that the low pH and the high heavy metal concentration of mine water contributed to the *Ktedonobacteria* and *Actinobacteria* natural selection and enrichment was further confirmed with the cannocial correspondence analysis (Fig. S2). The representative of leachate groups (MB1, MBS18) cluster closely with the low pH, high heavy metal concentration and *Ktedonobacteria* abundance as compared to the spring water samples which showed that after mixing with the adit leachate, the major abundant taxa *Proteobacteria* (MBS1-2) was replaced with *Actinobacteria* (MBS3).







Figure 5. Alpha and beta diversity. Shannon diversity index (a) and boxplots (b) depict that the spring water samples have the highest alpha diversity. The PCoA 2d (c) and 3d (d) graphs indicated the unconsolidated rocks ( ● ) microbial communities share similarity with both the spring water ( ● ) and mine leachate ( ● ) samples microbiome.



### 3.4 Appearance of sampled biofilms and phylogenetic analysis

Eukaryotic microalgae and aerial mycelia could be observed in light microscopy images when the collected leachate samples group was visualized under the light microscope (Fig. 6). Interestingly, a nematode related to *Poikilolaimus oxycercus* was also found which colonizes the deep subsurface sites (Borgonie et al., 2019) and one unicellular alga (*Coccomyxa subellipsoidea*) was highly abundant (unpublished previous data). TEM micrographs also showed eukaryotic (algal) along with prokaryotic cells in the biofilm, indicating cohabitation. Mineral deposition around the microbial cell walls was not observed, suggesting the inhabiting microbiota has employed some other pathways to cope with the heavy metal toxicity under low pH instead of metal precipitation (Fig. 6). TEM micrographs of the MB3 biofilm from the leachate group also revealed the presence of sporulating hyphae which could be identified as either mycelia-like branched *Ktedonobacteria*, or *Actinobacteria*. Since *Ktedonobacteria* being the most abundant class had much higher relative abundance as compared to the *Actinobacteria* and more importantly the distinguishing sporulation pattern of *Ktedonobacteria* (one spore per cell) ruled out the possibility of *Actinobacteria*.

Marsberg Kilianstollen offers a large reservoir of uncultured novel strains, comprising numerous *Ktedonobacteria* representatives (Fig. S3, S4). The genus level-based analysis led to the identification of at least 10 distinct uncultured genera affiliated with the *Ktedonobacteria* class and 80% of them could be classified to the *Ktedonobacteraceae* family. Within *Actinobacteria,* 20 known genera were identified, whereby the *Pseudonocardiaceae* family represented 88% actual abundance of these genera members. The most abundant genus, designated C0119, has been frequently assigned to *Ktedonobacteria* based on reference database comparisons (Dube et al., 2019; Paul Chowdhury et al., 2019). Nevertheless, C0119 OTU sequences have been shown to not cluster with *Ktedonobacteria* based upon phylogenetic analysis of *Chloroflexi* sequences (Glöckner, 2019; Jones et al., 2017) (Fig. S5). The unclassified strains might need to be classified into a new class of the cold climate adapted extremophiles belonging to the phylum *Chloroflexi*. In contrast, most of the *Actinobacteria* OTUs were classified as representatives of known genera such as the rare *Actinobacteria* (*Mycobacterium*, *Nocardioides*, *Micrococcus*, *Crossiella*, *Amycolatopsis*, *Nocardia* and *Pseudonocardia*) which could be potentially interesting with respect to novel bioactive compounds, biodegradation and biodeterioration pathways [28-30].





Figure 6. Light and transmission electron microscopy of Marsberg Kilianstollen leachate group biofilms. The images (a-c) captured under the light microscope depict the aerial mycelium and green algal biomass along with the white bacterial biofilm. TEM micrographs (d) and (e) show the mycelial growth and spores formed at short branches along a hypha resembling *Ktedonobacteria* sporulation pattern (hypha and exospore are marked in d) and the large eukaryotic algal cells (e), respectively.

## 3.5 Key metabolic pathways to survive under extreme conditions

### 3.5.1 Metagenome-assembled genomes (MAGs)

Biofilm sample MB1 as a representative of leachate group was selected for shotgun metagenomic sequencing based on the abundance of overall representative taxa of interest such as *Chloroflexi* and *Actinobacteria*. Eight relatively complete MAGs (completeness, ≥89%) were obtained, and subsequently proteins and metabolic pathways predicted. The genome sizes of the





eight MAGs (designated Mberg 002, 006, 008, 009, 010, 011, 015, and 019), showing a contamination rate ≤10%, ranged from 2.6 to 4.9 Mb. Furthermore, the number of identified genes ranged from 2516 to 4772. Based on phylogenetic analysis the MAGs were assigned to *Actinobacteria, Binatia, Deinococci, Chloroflexota, Dehalococcoidia, Chloroflexia and Ktedonobacteria* (Fig. S6).

### 3.5.2 Orthologous gene clusters

The preliminary comparison and analysis of GOSlim terms for core orthologous gene clusters for each MAG revealed unique survival pathways involved in extremophily (Fig. 7). The identified protein genes were grouped in 5540 distinct, 5395 orthologous and unique 145 single-copy gene clusters. The shared orthologous gene ontology GO terms correspond to common cellular functions such as respiration and cell wall synthesis. These orthologous clusters contribute to various survival pathways such as aromatic and sulphur compounds metabolic processes, detoxification pathways for arsenic compounds and heavy metals ions transport, dehalogenation, and sporulation. The detected broad-spectrum heavy metal binding domains and associated proteins identified in the PROKKA annotations are also important to survive high concentrations of heavy metals in the environment. The metal binding proteins are mostly enzymes which require transition metals Zn, Cu, Co, Ni as cofactor to perform different biological processes. The cupredoxins superfamily proteins and domains were frequently found in shared clusters and these contain type 1 Cu binding sites which are involved in oxidation reactions conferring resistance against Cu by various proteins (azurin, multicopper oxidases [MCO], laccases, and nitrosocyanin) (Arguello et al., 2013; Vita et al., 2016; Donaire et al., 2002; Redinbo et al., 1994; Zaballa et al., 2012). Mberg 010, assigned to *Binatia*, showed orthologous protein clusters related to aromatic compound degradation pathways. Mberg019, affiliated with *Ktedonobacteria*, encodes unique plasma membrane proteins, identified in unshared clusters. All MAGs seem to mediate transition metal homeostasis as metal binding proteins and transporters were frequently identified. The process of spore formation seems to be prevalent in all MAGs as GO terms for sporulation were observed in shared and unshared orthologous clusters. To study the MAGs with respect to heavy metal homeostasis, efflux systems, as well as detoxification and aromatic degradation pathways in more detail, pathway maps (https://doi.org/10.25625/W9PWCX) were generated based on PROKKA outputs and subsequently inspected.



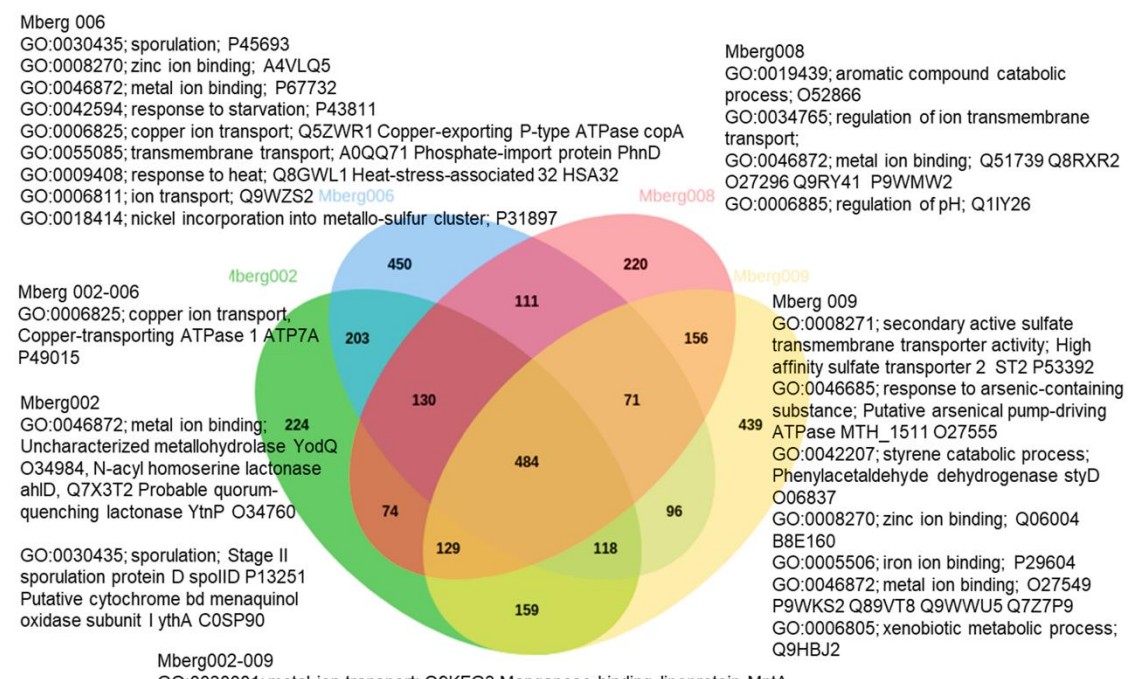

Mberg 006
GO:0030435; sporulation; P45693
GO:0008270; zinc ion binding; A4VLQ5
GO:0046872; metal ion binding; P67732
GO:0042594; response to starvation; P43811
GO:0006825; copper ion transport; Q5ZWR1 Copper-exporting P-type ATPase copA
GO:0055085; transmembrane transport; A0QQ71 Phosphate-import protein PhnD
GO:0009408; response to heat; Q8GWL1 Heat-stress-associated 32 HSA32
GO:0006811; ion transport; Q9WZS2
GO:0018414; nickel incorporation into metallo-sulfur cluster; P31897

Mberg 002-006
GO:0006825; copper ion transport, Copper-transporting ATPase 1 ATP7A P49015

Mberg002
GO:0046872; metal ion binding; 224 Uncharacterized metallohydrolase YodQ O34984, N-acyl homoserine lactonase ahlD, Q7X3T2 Probable quorum-quenching lactonase YtnP O34760

GO:0030435; sporulation; Stage II sporulation protein D spoIID P13251 Putative cytochrome bd menaquinol oxidase subunit I ythA C0SP90

Mberg008
GO:0019439; aromatic compound catabolic process; O52866
GO:0034765; regulation of ion transmembrane transport;
GO:0046872; metal ion binding; Q51739 Q8RXR2 O27296 Q9RY41 P9WMW2
GO:0006885; regulation of pH; Q1IY26

Mberg 009
GO:0008271; secondary active sulfate transmembrane transporter activity; High affinity sulfate transporter 2 ST2 P53392
GO:0046685; response to arsenic-containing substance; Putative arsenical pump-driving ATPase MTH_1511 O27555
GO:0042207; styrene catabolic process; Phenylacetaldehyde dehydrogenase styD O06837
GO:0008270; zinc ion binding; Q06004 B8E160
GO:0005506; iron ion binding; P29604
GO:0046872; metal ion binding; O27549 P9WKS2 Q89VT8 Q9WWU5 Q7Z7P9
GO:0006805; xenobiotic metabolic process; Q9HBJ2

Mberg002-009
GO:0030001; metal ion transport; Q9KFG3 Manganese-binding lipoprotein MntA

Mberg 006-008 GO:0009651; response to salt stress, Q9ZW31

All Mberg GO:0046914; transition metal ion binding; P0A673

Mberg011
GO:0022857; transmembrane transporter activity; Q58026
GO:0005506; iron ion binding; P29604 Ferredoxin
GO:0030435; sporulation; C0SP90

Mberg010
GO:0019439; aromatic compound catabolic process; Q51975 Q51974 O33477 O85673 O52379 Q53122 A5W4G5
GO:0046232; carbazole catabolic process; Q8G8B6 Q8GI14
GO:0019383;(+)-camphor catabolic process; H3JQW0
GO:0015673; silver ion transport; P38054 Cation efflux system protein CusA
GO:0046686; response to cadmium ion; P94177 Cation efflux system protein CzcA
GO:1901359; tungstate binding; O57890 Molybdate/tungstate-binding protein WtpA
GO:0018896; dibenzothiophene catabolic process; P54998
GO:1901170; naphthalene catabolic process; Q9X9Q6
GO:0010128; benzoate catabolic process via CoA ligation; Q51601
GO:0043640; benzoate catabolic process via hydroxylation P19076 P07770
GO:0042203; toluene catabolic process P49155
GO:0006790; sulfur compound metabolic process D3RPB9
GO:0019380; 3-phenylpropionate catabolic process; P0ABR6
GO:0046687; response to chromate; P17550
GO:0046872; metal ion binding; E7FHP1 P00200 Q89VT8 O34760 Q7X3T2
GO:0004497; monooxygenase activity; G2IN04

Mberg015
GO:0055085; transmembrane transport; P9WG00 Q6CZ31 P77156 P37624 Q9KEE9 O51924 Q57RB0 P39642 Q66FU7 O32155 O07017 P46904 Q7CH99 P21409 P9WG00 O32155 P40980 O51924 P40980 O32155 O58760 P55603 P06109
GO:0046872; metal ion binding; Q2YTD2 O34984 Q9RY41 P9WMW2 Q5SME3
GO:0030435; sporulation P13251 P45693
GO:0055072; iron ion homeostasis; P37009
GO:0016784; 3-mercaptopyruvate sulfurtransferase activity; Q9I452
GO:0008270; zinc ion binding; Q06004 A4VLQ5
GO:0006885; regulation of pH; Q1IY26

Mberg019
GO:0030435; sporulation Q9K5N0 B6H9U8
GO:0018414; nickel incorporation into metallo-sulfur cluster; P31897
GO:0008270; zinc ion binding; Q06004
GO:0046872; metal ion binding; O27549 O27296 P67732 Q4L749
GO:0019439; aromatic compound catabolic process; O52866 GO:0006825; copper ion transport; Q5ZWR1 Copper-exporting P-type ATPase CopA
GO:0034765; regulation of ion transmembrane transport; Q8RXR2
GO:0009432; SOS response; Q5L0C2
GO:0005886; plasma membrane; O31603 A6T022 A6T022 P12752 A3DC75 O34578 Q55705





Figure 7. Venn diagram displaying the distribution of shared orthologous clusters among the eight assembled genomes. The orthologous clusters of similar proteins have been allocated the Go terms based on same function or process. Only the Go terms related to metals, aromatic compound, sporulation, and pH regulations are mentioned to identify the microbial survival under extreme stress conditions. The closely matched proteins are identified as Swiss-prot accession numbers and names of some important hits are mentioned.

### 3.5.3 Heavy metal homeostasis and efflux systems

Transporters such as copper-exporting P-type and oxidation enzymes involved in transition metals homeostasis were identified in all selected MAGs (Fig. 8). The genes involved in copper homeostasis include copper-sensing transcriptional repressors CsoR and RicR, copper-exporting P-type ATPases CptA, ActP and CopA, and oxidation enzymes, multicopper oxidases (MCOs) in all MAGs. Basically, upon Cu(I) ion binding, the dissociation of CueR, CsoR and RicR transcriptional repressors (Fu et al., 2014; Smaldone and Helmann, 2007) activates the copper regulon genes related to transporters CopA-like P-type ATPases, metallothioneins or copper binding chaperons CopC, CopZ and periplasmic multicopper oxidase MmcO, CueO (Osman et al., 2010; Shi et al., 2014). Under anaerobic conditions the multicopper oxidase enzymes become inactive, and therefore another three-component channel/pore Cus complex controls Cu(I) efflux through CusA, an inner membrane energy-driving channel which is attached to the outer membrane pore CusC through the periplasmic CusB protein (Outten et al.).

The *cutA* locus, presumably involved in copper tolerance and homeostasis, was also characterized to affect tolerance levels to zinc, nickel, cobalt and cadmium ions (Fong et al., 1995). The metalloregulatory transcriptional response to di- and multivalent heavy metal ions Cd(II), Pb(II), Bi(III), Zn(II) as well as Cu(II) is maintained by SmtB/ArsR family repressors CmtR and CadC (Busenlehner et al., 2003). MntH is a divalent metal cation transporter which displays broad substrate specificity and can regulate the intracellular accumulation of several divalent cations, including Mn(II), Cd(II), Co(II), Zn(II) and, to a lesser extent, Ni(II) and Cu(II) (Makui et al., 2000). CzcD, a heavy metal cation efflux transporter, mediates heavy metal resistance with respect to Cd(II)/Co(II)/Zn(II) in the absence of the high resistance CzcCBA system (Nies, 2003; Anton et al., 1999; Papp-Wallace and Maguire, 2006).

Manganese homeostasis is maintained by a manganese efflux pump, MntP (Waters et al., 2011). A putative bacterial multicopper oxidase MoxA has been reported to compact the cellular Mn(II) toxicity through surface oxidation to the insoluble Mn(III) and Mn(IV) oxides (Ridge et al., 2007; Zhang et al., 2015). HoxN, a high-affinity nickel transporter facilitates the nickel translocation process as nickel permease (Wolfram et al., 1995). The influx of Mg(II), Ni(II) and Co(II) is coordinated via an ubiquitous divalent metal ion transporter CorA (Kersey et al., 2012) and efflux of these divalent metal ions is directed through CorC, a magnesium/cobalt efflux protein (Gibson et al., 1991). The homeostasis of the transition metals cobalt, nickel and iron and Ni(II) and Co(II) detoxification is also regulated by a nickel-cobalt exporter, designated RcnA, through efflux (Koch et al., 2007).



Cellular zinc uptake is regulated through the energy intensive import system ZnuABC, where ZnuA binds Zn(II) in the
periplasmic space and docks Zn(II) into the membrane permease ZnuB and the ZnuC of the pump finally catalyses ATP-
dependent Zn(II) import into the cytosolic environment. In contrast, ZupT is involved in less energy intensive non-specific
Zn(II) uptake along with the transition metals Fe(II), Co(II), Cd(II), and Mn(II) along the concentration gradient. Under high
cytosolic divalent Zn(II), Cd(II), and Pb(II) concentrations, the MerR homologue, ZntR induces a Zn(II)-, Cd(II)-, and
Pb(II)-transporting P-type ATPase ZntA (Rensing and Mitra, 2007).

The oxyanions molybdate and tungstate are taken up through the membrane by the high-affinity ModABC molybdate system
along with sulphate ions (Markovich, 2001). Sulphate and thiosulphate are taken up by sulphate permeases, carriers
belonging to the SulT family, encoded by the cysPTWA operon, and SulP family members (inorganic anion uptake carriers)
(Kertesz, 2001).



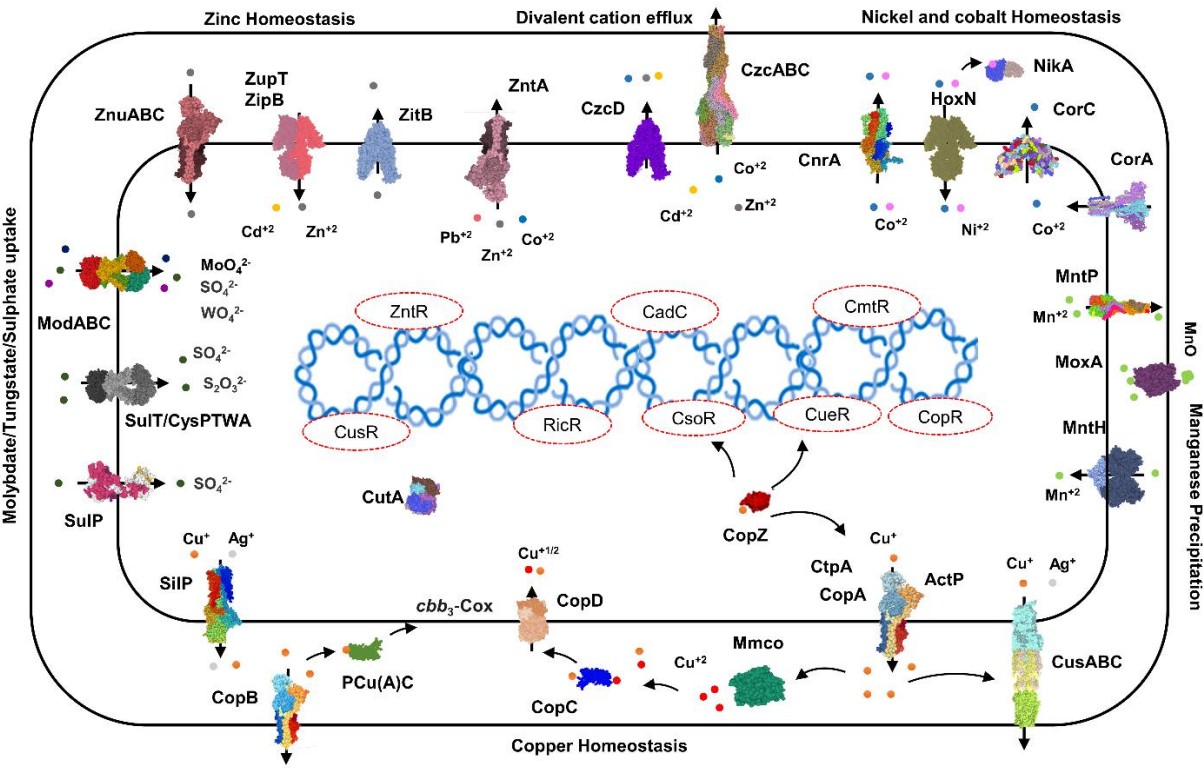


Figure 8. Heavy metals transport channels and enzymes. The metalloregulatory transporters, enzymes, chaperones, and transcription factors form the specific protein-metal coordination complexes involved in the heavy metal transport, intracellular trafficking, storage, and detoxification are derived from the MAGs pathway maps (supplementary pdf files) and Prokka annotations (Arif et al., 2021a). Copper homeostasis is maintained through the efflux pumps CopA, CtpA, ActP and

CopB under aerobic conditions which remediate the high cellular Cu(I) contents. The CopZ acts as allosteric switch to detect the excess copper ions and activate the transcription of the CopABCD operon and multicopper oxidase MmcO through Cu-based inactivation of CsoR and CueR repressors. The periplasmic Cu(I) is oxidized to the less soluble Cu(II) by MmcO and CopC that both bind Cu(I/II) as storage and metallochaperone proteins, while PCu(A)C chaperones facilitate the biogenesis of the copper center in the cytochrome oxidase.




### 3.5.4 Detoxification and aromatic degradation pathways

To successfully colonize the acid mine leachate downstream sites, microbes should harbour acid resistance, detoxification, and metabolizing systems as identified in all MAGs (Fig. 9). Arginine-dependent acid resistance, encoded by the MAGs, relies on arginine decarboxylase SpeA, which decarboxylates arginine (Arg) to produce agmatine (Tsai and Miller, 2013;

Richard and Foster, 2004). Mercuric reductase MerB reduces divalent Hg(II) to volatile mercury Hg (Silver and Hobman, 2007) and arsenate reductase ArsC reduces the arsenate As(V) to arsenite As(III) which is delivered to ATP-dependent anion pump ArsAB by the metallochaperone ArsD (Martin et al., 2001). The enzymes related to aromatic compounds; chlorinated phenols, benzoate, atrazine, cinnamate, biphenyl, phenylacetate carbazol, catechol and 4-sulphocatechol phenylethylamine, naphthalene and 5-nitroanthranilate degradation were mainly identified as hydroxylases, dioxygenases, dehydrogenases,

epoxidase etc. To detoxify reactive oxygen species (ROS), toxins, and antibiotics compounds with thiols, mycobacteria and some other *Actinomycetales* utilize mycothiol-mediated detoxification through Mca enzyme (Newton et al., 2008; Newton et al., 2000). Other detected important enzymes related to ROS detoxification were superoxide dismutase SOD and peroxidases (Broxton and Culotta, 2016).





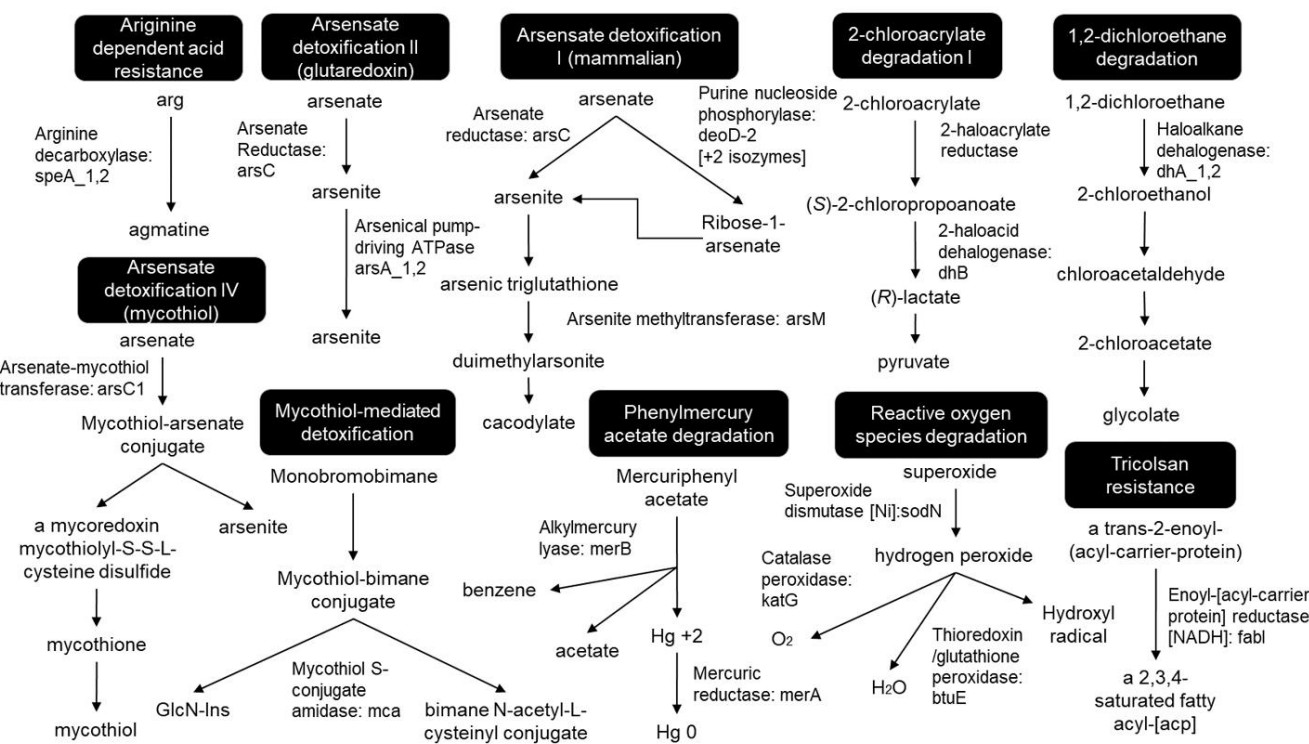


Figure 9. Detoxification pathways. The pathways have been redrawn from the pathway maps of all MAGs, provided in the supplementary pdf files.

## 4. Discussion

Generally, the habitats, which exceed most natural heavy metal concentrations, are dominated by prokaryotes with chemolithoautotrophic lifestyles, which produce sulphuric acid and are responsible for the mobilization of heavy metals from rocks through the oxidation of metal sulphide (Schippers and Sand, 1999). In accordance with this, a significant heavy metal mobilization of Cu, Zn, Mn, Pb, and Ni from copper slag debris has been already measured with enrichment cultures of iron/sulphur-oxidizing *Acidithiobacillus ferrivorans* and *Leptospirillum* obtained from the Marsberg Kilianstollen
iron/manganese sampled crusts (Amin et al., 2018). However, the present study aims to give a complete microbiome overview of Kilianstollen habitats influenced by mine drainage, comprising leachate downstream colonization sites, where





we could expect microbial metal reduction, deposition, and detoxification. To colonize the Marsberg mine subsurface habitat, the microbes must have developed resistance against heavy metals such as molybdenum, manganese, cobalt, zinc, and copper. Above optimal concentrations, copper confers its toxicity to microbes through its redox activity which catalyses

a Fenton-like reaction, resulting in the generation of reactive oxygen species that may cause protein damage and lipid peroxidation (Macomber and Imlay, 2009) and destabilizing the iron-sulphur clusters via Fe(II) displacement in key enzymes prosthetic groups (Azzouzi et al., 2013; Chillappagari et al., 2010; Dupont et al., 2011). Thus, copper may inhibit the growth of common fast-growing *Proteobacteria* or *Firmicutes* such as *Escherichia coli, Pseudomonas aeruginosa*, *Vibrio spp*., *Bacillus cereus*, and *Bacillus subtilis*, even at micromolar concentrations (Gordon et al., 1994). This metagenomic study of

the Marsberg copper mine also observed the colonization of heavy metal tolerant microbes and inhibition of *Proteobacteria* and *Firmicutes* growth under the influence of copper toxicity. The colonization sites near the copper-rich leachate outflow stream such as a copper flume (MB1-6 biofilms) enrich copper resistant groups belonging to abundant phyla *Chloroflexi* (*Ktedonobacteria*), *Actinobacteria* and *Cyanobacteria* (*Oxyophotobacteria*) as compared to the freshwater stream sites; mainly colonized by fastidious non-resistant *Proteobacteria*. A metatranscriptomic study of an abandoned Pb-Zn mine (Coto

Txomin, Spain) also determined that the heavy metal concentrations (up to 3220 and 97 g kg$^{-1}$ of Pb, Cd, and Zn, respectively) and low pH (4-6) drastically influenced the soil microbial diversity, suppressed the relative abundance of *Actinobacteria*, *Acidobacteria*, and *Alphaproteobacteria*, and enhanced slow-growing metal and acid-tolerant taxa affiliated with *Chloroflexi* (*Ktedonobacteria*) (Epelde et al., 2015).

The same enrichment trend for *Chloroflexi* (*Ktedonobacteria* and KD4-96) was observed when a microcosm setup

supplemented with acid mine drainage contaminated soil and cysteine hydrochloride was incubated for 6 months at 30 °C, which decreased the abundance of the major taxa *Acidobacteria*, *Acidimicrobiia, Actinobacteria*, and *Thermoleophilia* (Gupta et al., 2018). A high abundance of *Ktedonobacteria* at the downstream arsenic deposits of the acid sulphate hot spring Tengchong, China, suggests that these aerobic heterotrophic mesophiles and thermophiles may have been involved in arsenic reduction/toleration along with iron and sulphur oxidation cycles (Jiang et al., 2016). The Marsberg cold-adapted

*Ktedonobacteria* also colonized the downstream acid mine leachate, indicating the ability to reduce heavy metal ions or tolerate the sediments along with sulphur compounds oxidation. The co-occurrence of *Cyanobacteria* (*Oxyphotobacteria*) and *Chloroflexi* (*Ktedonobacteria*) could facilitate the growth of heterotrophs by providing carbon nutrients and in return the heterotrophs may remediate the heavy metal contaminated sites, resulting in better microbial survival and colonization of the microbial consortium. The selective enrichment of *Ktedonobacteria* at Marsberg copper mine indicates the ability of these

psychrophiles to inhabit cold environments and could be linked to heavy metal tolerance along with iron and sulphur oxidation cycles. The extreme conditions survival is attributed to the high metabolic plasticity of *Ktedonobacteria,* a diverse class ranging from thermophilic to mesophilic isolates*,* and the type stain *Ktedonobacter racemifer* has an unusually large genome of 13 Mbp, containing 9539 genes, 601 of which are transposases (Chang et al., 2011). The identified KO07665 based on KEGG analysis belonging to the *Ktedonobacteriaceae* codes for the copper resistance phosphate regulon response

regulator CusR (Thomas Iv et al., 2020). The pathway map and annotation of the Marsberg *Ktedonobacteria* MAG also



indicates a detection mechanism with respect to Cu(I) by copper-sensing transcriptional repressor (RicR), oxidation to Cu(II) via multicopper oxidase (MCO) and finally export outside the cells through copper-exporting P-type ATPases (CopA, CtpA) and non-specific heavy metal cadmium, cobalt and zinc/H(+)-K(+) antiporter (CzcD).

High concentrations of heavy metal ions in the environment promotes selection for heavy metal-resistant microbes, with either chromosomal or plasmid-level genes, maintaining heavy metal ion homeostasis inside cells (Rademacher and Masepohl, 2012). The organisms may reduce the sensitivity by employing the permeability barriers, by enhancement of active transport of metal ions from the cytoplasm (efflux) through a specific membrane transport system, by enzymatic detoxification, by reduction of metal ions by redox reactions, or by complexation of heavy metals resulting in extracellular and intracellular sequestration, etc. (Silver and Phung, 2005; Hobman et al., 2007). For all organisms in heavy metal 475   environments, one or more of these adaptations are prerequisites for survival. This study investigated the pathways crucial for microbial survival nearby acid mine drainage, and all MAGs explicated specialized adaptations to cope with heavy metal toxicity. Especially copper homeostasis is maintained by copper-exporting P-type ATPases efflux pumps ActP, CptA and CopA (Arguello et al., 2013; Kim et al., 2008; Festa and Thiele, 2011), Cu(I)-specific metalloregulatory proteins (CsoR, RicR (Fu et al., 2014) and CopR (Villafane et al., 2011)), resistance to acidity mediated copper toxicity by ActP (Reeve et 480   al., 2002), oxidation of Cu(I) to less toxic Cu(II) by the cupredoxins proteins superfamily such as periplasmic multicopper oxidases (MCOs) (Rowland and Niederweis, 2013) and expression of metallochaperones and metallothioneins with binding constants for Cu in the picomolar–femtomolar range (Rae et al., 1999; González-Guerrero and Argüello, 2008). The sensory cytosolic CopZ-like chaperones stimulate the transcription of several copper-stress related genes via trafficking the Cu(I) to the DNA-bound CsoR and CueR and the inner membrane-localized channels, especially copper-exporting P-type ATPases 485   such as ActP, CptA and CopA for efflux to the extracytosolic periplasmic environment (Arguello et al., 2013; Kim et al., 2008; Festa and Thiele, 2011; Novoa-Aponte et al., 2019). Mounting evidence also confirms the coexistence of the anaerobic Cus system with the aerobic CopA regulon in 44% of γ-proteobacteria group members (Hernández-Montes et al., 2012). The periplasmic CopC proteins maintain the bacterial copper homeostasis via binding both Cu(I) and Cu(II) along with CopD inner membrane protein (Cha and Cooksey, 1993). PCuAc-like chaperone PccA  required for the biogenesis of the copper 490   centre assembly in the cbb3-type cytochrome c oxidases (CcO) have been demonstrated in various bacteria (Thompson et al., 2012; Andrei et al., 2020). CopA has also been shown to compete for Cu(I) with sub-femtomolar affinity against copper binding ligands and the efflux rate of this enzymes has been measured up to 27-130 nmolmg$^{-1}$mn$^{-1}$ (Wijekoon et al., 2017). Deletion of efflux channels render the cells susceptible to copper toxicity via intracellular accumulation of heavy metals (Sitthisak et al., 2007). The intracellular copper and zinc ($10^{-6}$ and $10^{-4}$ M, correspondingly) is buffered at a bound state as 495   their transcriptional regulators CueR, Zur, and ZntR bind the cognate metals with very high affinities ($10^{-21}$ and $10^{-15}$ M). This indicates that the free copper and zinc cytoplasmic concentrations are to be closely regulated to avoid toxicity as compared to manganese and iron (low regulators affinities $10^{-5}$ M and $10^{-6}$ M, respectively) (Porcheron et al., 2013). Additionally, manganese homeostasis is maintained by a manganese efflux pump (MntP), as Mn is essential for enzymatic





catalysis in the carbon metabolism, extracellular capsule polysaccharide synthesis and protection against oxidative stress
(Waters et al., 2011).

Since most of the heavy metal efflux pumps are ATPases, a constant supply of energy (as ATP) is met through converting various aromatic compounds to TCA cycle intermediates (Fuchs et al., 2011; Ghosal et al., 2016), in addition to several inorganic nutrients catabolism. To detoxify ROS, toxins, and antibiotic compounds, mycobacteria and some other Actinomycetales utilize mycothiol-mediated detoxification (Newton et al., 2008; Newton et al., 2000). Other important
enzymes related to ROS detoxification were superoxide dismutase SOD and peroxidases which degrade the superoxide anion radicals (Broxton and Culotta, 2016). The *arsRDABC* operon codes for an ATP-dependent anion pump that confers resistance to arsenite, arsenate, antimonite, and tellurite (Rosen, 2002). Arsenite methyltransferase ArsM catalyses the formation of several volatile methylated intermediates from As(III), which eventually results in loss of arsenic from the cells through passive diffusion (Huang et al., 2018; Zeng et al., 2018). The mercuric reductase MerB reduces divalent Hg(II) to
less bioavailable metallic mercury $Hg^0$ vapour which is also volatilized under aerobic conditions and leaves the cells through passive diffusion (Silver and Hobman, 2007). These detoxification pathways also explain the lack of mineral or metals deposition as visible in TEM images as either oxidation of metallic ions to less soluble ionic form or their reduction to volatile products makes them leave the bacteria through extrusion or diffusion under low pH. Hence, to inhabit the Marsberg acid mine leachate sites, the cold-adapted microbiome has the plasticity to express a wide range of heavy metal-specific
enzymes which could either oxidize or reduce different metallic ions to neutralize their toxic effects, regulate their intracellular concentrations and integrate the heavy metals as cellular components, cofactors for enzymatic functions, protection against oxidative stress. Culturing of specific strains especially *Ktedonobacteria* may further clarify mechanisms of heavy metal resistance (exporter systems, metal chelators, bioorganic compounds enhancing metal precipitation). This might also give implications to mine waste treatment, bioremediation and biomining.

**5. Conclusion**

The freshly collected samples from Marsberg copper mine (NE Rhenish Massif, Germany) were taken as a hitherto unexplored inventory of extremophilic organisms with largely unknown properties with respect to long-term survival, heavy metal tolerance, and degradation of complex organic compounds. The typical colonization patterns, mainly composed of *Firmicutes,* and *Proteobacteria* changed considerably towards uncultured *Chloroflexi*, including *Ktedonobacteria*
representatives, when the sampling sites around the spring water stream shifted to the acid mine leachate outflow. The acid mine drainage with influx of heavy metals altered the composition, drastically reduced the richness and evenness of microbial communities, and exerted selective pressure towards resistance to metal contamination. Consequently, the microbiome has evolved various survival pathways related to aromatic and sulphur compounds metabolism, toxic arsenic compounds reduction, copper ions oxidation and heavy metal ions reduction and extrusion, dehalogenation and sporulation.



## 6. Acknowledgements

The support of Petra Ackermann, Gerhard Rosenkranz and the "Marsberger Heimatbund e. V." (Marsberg) is gratefully acknowledged. We would like to thank Deutscher Akademischer Austauschdienst (DAAD) for the provision of a doctoral research grant. The support by the Open Access Publication Funds of the University of Göttingen is also acknowledged. The authors declare no conflict of interest.

## 7. Contributions

Conceptualization, M.H., H.N., and S.A; Methodology, S.A., M.H., E.S., A.R., G.A., H.N.; Sample collection; M.H., S.A., E.S.; Formal analysis; S.A., E.S., A.R., G.A., H.N.; Writing—original draft, S.A., G.A., H.N..; Writing—review and editing, M.H., S.A., H.N., A.R., and G.A..; Funding acquisition, M.H, A.R. and S.A.

## 8. Competing interests

The authors declare no competing interests

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
