# Peer review of "Composition and Niche-Specific Characteristics of Microbial Consortia Colonizing Marsberg Copper Mine in the Rhenish Massif"

_Biogeosciences, 2021_

## Author Response (AR1)

The manuscript by Arif et al. sought to characterize microbial communities in mine drainage of the Marsburg Copper Mine. Waters in the copper mine span a gradient of heavy metal concentrations with samples collected in the copper precipitation flume having mg/L quantities of heavy metals. Microbial communities were characterized in all samples collected and these samples grouped into leachate, spring water, and unconsolidated rocks. The various samples were analyzed to look at the gradient in chemistry and distribution of microbial taxa. The authors selected a single leachate sample for metagenomic sequencing which yielded 8 MAGs. The combination of amplicon-based and metagenomic sequencing was a nice strategy as the MAGs gave the ability to assess potential metabolic function better than 16S rRNA gene sequencing alone. The authors found that members of the Actinobacteria and Chloroflexi were numerically dominant and that the MAGs contained a number of heavy metal survival mechanisms. Overall, this was a really interesting study with a good approach. However, I have a number of suggestions listed below to improve clarity. I also encourage the authors to consider additional comparison of their results to other acid mine drainage systems.

The authors would like to thank referee for her constructive suggestions and comments

1. 14 and elsewhere: please use caution with the term "metagenome" when referring to your 16S rRNA gene amplicon data. Although, metagenomics in its strictest definition includes amplicon sequencing the prevailing use of the term in microbiology is in reference to genomic sequencing of multiple organisms in a single community. Change this throughout the paper to something like community amplicon sequencing. This change will also highlight the fact that you performed both metagenomics and amplicon-sequence based community characterization for your study.

Separated the amplicon sequencing from metagenomics in the manuscript.

2. 51 and l. 65: the concept of "cold" is mentioned here but not expanded upon or highlighted as a focus of the study until the discussion and conclusions. Without more explanation I don't see the justification for this to be a unique aspect of the site and why it would be novel at all. Why is studying a cold environment with heavy metal enriched waters important? How is "cold" quantified and relative to what? Comparisons to other heavy metal-contaminated sites and their temperatures/microbial communities would be valuable here.

Since, Marsberg Kilian Coppermine has a constant temperature of 10 °C, it is referred as cold environment. It was not expanded because 10 °C is not a good borderline between psychrophiles and mesophiles as both could grow on it. *Ktedonobacteria* cultured members are either mesophiles or thermophiles but psychrophiles are reported through amplicon sequencing only. In the scope of this study, it was not possible to culture *Ktedonobacteria* to validate its temperature range and particular role in a cold heavy-metal comminated site.

78: please provide (throughout the paper) the rational for performing metagenomics on a single sample and especially why this particular sample was selected.

Explained in line 83 that the MB1 biofilm was selected because of the high abundance of *Chloroflexi* (*Ktedonobacteria*) and line 225-229: *Extracted DNA from one of the leachate biofilm samples, MB1 abundant in Ktedonobacteria (see fig. 4) as a representative of*

*leachate group was submitted to the Göttingen Genomics Laboratory for shotgun metagenomic sequencing. The rationale behind selecting MB1 biofilm was to investigate the survival mechanisms that contributed to the high abundance of Ktedonobacteria around the toxic copper-rich leachate stream.*

188-189: this sentence, especially the phrase "as a representative of leachate group" is not very clear. Can this sentence be revised for clarity? It might be valuable to provide some more detail on the sample selection here or to refer the reader to the results where community data is used to justify selection of MB1.

corrected

3. 323-324: "MB1 as a representative of the leachate group" – looking at Figure 4 MB1 has a number of differences compared to the other leachate group samples. What was "representative" based upon? It actually has a lower abundance of Actinobacteria compared to the other leachate samples, while Chloroflexi are in higher abundance than 4 of the leachate samples. Regardless of how/why this sample was chosen please add details on the reasoning and selection process so that it is clear to the reader how the MAGs fit into our understanding of the overall community in the system.

Since *Ktedonobacteria* were abundant in leachate biofilms and mostly members of *Ktedonobacteria* are uncultured and unknown, we were interested to investigate its heavy metal resistance and aromatic compounds metabolism. That's why MB1 was selected as it has more *Chloroflexi(Ktedonobacteria)* reads. Moreover, it was possible from this sample to extract enough DNA for metagenomic sequencing.

4. 91: I think you are missing a "respectively" here

Corrected

5. Figure 1: can you add a symbol for the water table depth? It is unclear if the water depth is shown in the cross section or if the water table is being pumped down for mining activities

The water table depth of the streams is just few centimetres deep and wide. These narrow water streams are naturally flowing downwards and, in the NE, or SW directions without being pumped artificially.

6. 85: it is unclear to me if the "spring water" samples are meant to represent background, uncontaminated samples. The geochemistry suggest that they are uncontaminated from the mining activities but it is not easy to discern from Figure 1. Explicitly stating whether those are meant to be background samples would be valuable. Okay, I just re-read Section 3.1 and there it states that these are sources of fresh groundwater. I still suggest adding that detail to the methods.

Corrected in line number 94. The section 2.1 is rewritten.

7. 102: I would change this to genomic DNA to encompass both sequencing approaches used. (see comment 1 above)

Corrected

8. Figure 2: please add scale bars to all panels.

Corrected

9. 115-117: this sentence needs clarification to state which sample types were stored in which bottle type.

Corrected and explained in the manuscript (Line 144-146)

10. 122 and elsewhere: be consistent with use of charges for all anions and cations. Throughout the paper you switch between including or not including charges for elements. I usually only use charges for ions and not elements; make sure superscripts and subscripts for sulfate and nitrate are correctly formatted (e.g., L. 215 and 217; Figure 3 legend).

Corrected. The charges for the ions have been skipped to avoid confusion because we are talking about analysed total concentrations, not different specifically charged ions. With ICP-OES or ICP-MS we measured total amount of elements, e.g. Sr or Cu, not specific ions. Therefore, in tableS3 no charges are given, in fact everything is total concentration of dissolved element or parameter.

11. 135 and 149: omit "photometrically" as it is not necessary nor is it the best word choice.

Deleted. (155 and 170 line)

12. 202: the abbreviation was already defined

Corrected

13. Section 3.1: the methods state that PHREEQC was used to look "calculation of ion activities, pCO2 (partial pressure of CO2) of samples and mineral saturation states" but no data are presented. Please include those results or amend the methods section.

Modified. Supplementary information Excel Table S3 is uploaded at the Göttingen Research Online Database and unnecessary parts were deleted, which includes the whole calculation as mmol/l, calculated charge balance, saturation states and some minor important parameters (e.g. Cs, $PO_4$, silica etc.). Table 3 includes basics (pH etc.) and the analytic values as mg/L or µg/L .

14. 216-218 and elsewhere: make sure the units are written correctly with superscripts and spacing; check figures too.

Corrected

15. Figure 3: are the samples presented in the direction of water flow? Consider plotting the copper flume leachate heavy metal concentrations in mg/L.

Information added about the direction of water flow in the figure 3 captions. Concentration plotted in mg/L.

16. 243-244 and 254-257: It is not clear what data are being tested and compared here. Were the "actual abundances" (Fig. S3) or the "relative abundance" (Fig. 4) used in the statistical tests? What cutoff was used to designate what an "abundant taxa was"? I'm also wondering if an ANOVA is the best statistical analysis here. If you are only comparing relative abundance values, it doesn't take into account overall changes in total biomass. An analysis like DESeq2 or LEfSe would be more appropriate as it would take into account the differences in biomass and identify specific taxa that are changing in differential abundance.

The actual abundance values after normalization were used for ANOVA. As recommended by reviewer, DESeq2 has been applied for the differential abundant analysis of taxa at the phylum and class level and described in main text.

17. Figure 4: consider using the same colors in panels A and B for taxa that have the same name at both taxonomic levels. Check spelling in panel B for Oxyphotobacteria

Corrected

18. 271, 278-284 (and elsewhere): It is really important to be cautious with assigning causality with observational data. Your data didn't allow you to monitor enrichment or replacement of taxa so I would tone down the language and state these things more as hypotheses.

Corrected

19. 291-298: in this section it is not clear how the nematode and alga were identified to species level. Either here or in the methods a description of those methods would be valuable. Further, the identification of Actinobacteria hyphae to genus level is also not clear, although there is 16S rRNA gene data indicating that these organisms are highly abundant.

They were identified as a result of 18S amplicon sequencing. The highest frequency sequences matched 100% to several Chloropyta sequences according to the Blast search and when a species is given, most of them were identified as *Coccomyxa subellipsoidea*. Abundant actinobacterial genera and a phylogenetic tree are provided in the supplementary files.

20. Results: I found the order of presentation to be confusing, especially for section 3.4. Sections 3.2 and 3.3 present the 16S rRNA gene sequencing for community analysis then 3.4 jumps to microscopy then back to 16S rRNA gene-based phylogenetic analysis. It would be much easier to follow the story if the diversity analyses (section 3.3) were presented before distribution of taxa (section 3.2) with the microscopy and phylogenetic analysis (section 3.4) last. The jump from 16S rRNA gene data to microscopy would not be as severe as it seems that these data sets support one another.

Corrected and rearranged

21. 302-313: the phylogenetic analysis was based on 350 bp amplicon sequences which doesn't provide a lot of information for robust taxonomic affiliation or

phylogenetic inference. Making definitive statements about an OTU being rare or a novel class/species is a big reach based on limited sequence read length.

Corrected

22. 313: correct the reference style.

Corrected

23. 349: this is a great idea to publish these maps and make the details available to the reader.

Thank you

24. Figure 7: I find this figure hard to interpret. Is this really the best way to present these results? Could the figure be revised so that the text doesn't wrap or overlap the Venn diagram or the sample names? Caption: GO should be in all caps.

GO terms capitalized and simplified for readers.

25. 391: the topic of Mo and W uptake is of interest to me so I looked up the Markovich, 2001 reference. The ModABC molybdate system is not mentioned in this paper and the reference only assesses sulfate transporters in mammals. While Markovich does address membrane uptake of molybdate and tungstate along with sulfate I don't see how this is an appropriate reference for the ModAB system. This manuscript: https://www.frontiersin.org/articles/10.3389/fmicb.2018.03030/full, references a number of papers that would be more appropriate sources for that uptake system:

    321.    Self, W. T., Grunden, A. M., Hasona, A., and Shanmugam, K. T. (2001). Molybdate transport. Res. Microbiol. 152, 311–321. doi: 10.1016/S0923-2508(01)01202-5

    322.    Maupin-Furlow, J. A., Rosentel, J. K., Lee, J. H., Deppenmeier, U., Gunsalus, R. P., and Shanmugam, K. T. (1995). Genetic analysis of the modABCD (molybdate transport) operon of Escherichia coli. J. Bacteriol. 177, 4851–4856. doi: 10.1128/jb.177.17.4851-4856.1995

Corrected and added references.

26. Discussion: the discussion is highly focused on the MAG results and little about the overall system and community dynamics. Combining the results and discussion sections might be a stronger approach for the paper.

The biofilms details and their abundant taxa have already been briefly discussed in https://journals.asm.org/doi/full/10.1128/MRA.01315-20. Combining both sections at this stage is again a lot of work, therefore the overall community dynamics is being discussed in discussion section again

27. 427-429: the previous microbial work at the site would be a good addition to the introduction.

Added in line 75

28. 439: omit metagenomic here and use a broader term/phrase

Corrected in 486 line.

29. 474: omit "etc"

Corrected in 522 line .

30. 476: add "had" before specialized

Corrected in 525 line.

31. 477-481: this sentence is much too long and really difficult to follow.

Revised in line 527-530

32. 487: change to Deltaproteobacteria

Changed to Gammaproteobacteria as it is gamma symbol γ in line 535.

33. 492: correct the units

Deleted.

34. 493-500: these sentences are very confusing and it is unclear how these affinities fit with your metagenomic data sets. The information takes away from, instead of strengthening, your study.

Deleted.

35. All Figures: Please verify that all figures meet formatting standards to be color blind friendly. In many cases the only way to tell different data sets apart is via color (e.g. Figure 5). Using different symbols and colors would make your figures accessible to a larger audience.

Some figures were changed to black and white color.

36. Bar charts are challenging to format to be colorblind friendly. So, if you cannot find an appropriate color scheme consider including the data presented as a table in the supplemental information or as another dataset at the Göttingen Research Online Database.

The excel files are uploaded in the Göttingen Research Online Database.

37. References: check all references for correct formatting, capitalization, italics, and use of consistent journal names vs. abbreviations. Also, the final reference is out of order.

Corrected.

38. Figure S3: the y-axis and caption should be changed **to** number of reads instead of "actual abundance" since that describes the data presented more accurately.

Corrected Figure S3.

39. Figure S3, Inset pie charts: Do these show the overall abundance for all samples presented in the figure? Are these number of reads or relative abundance?

Yes, it is overall abundance for all samples and based on number of reads. This information is added in line 35 of supplementary file.

40. Figure S5: I suggest making panels A and B larger such that each tree fits on a whole page. Since this is supplemental information its fine to take up a lot of space.

Corrected in line 41 and 42.

Reviewer 2:

This study characterized the soil microbial diversity and metabolic potentials in the Kilianstollen Marsberg, a copper mining area in German. I have several concerns about this manuscript:

1. This study is just region research, how to attract global interests? Is there any implications for the other mining sites? Authors should highlight its "unique" in the Introduction.

The colonization of the similar *Chloroflexi* (*KD96, Ktedonobacteria* classes) is common at heavy metal contaminated sites and have been discussed in detail in the discussion section. The investigation of the *Chloroflexi, Cyanobacteria* and *Actinobacteria* stains may eventually be used to remediate the heavy metal contaminated sites.

2. Plz highlight the hypothesis. What is the scientific question in this study?

The last paragraph has been modified as per instructions from reviewer 1, the previous study has been added to highlight the hypothesis: to observe whether the mine waters enriched in transition metals may be toxic to microbial inhabitants or, conversely, support unique forms of metal respiration and enrich resistant microbial consortia under oligotrophic conditions.

3. Plz move the details of PCR to supporting information.

Corrected.

4. Just recommendation (Line 160): ASV is more widely acceptable than OTU.

Yes agreed, the study has already been commuted from OTUs.

5. Line 190: Provide the detail parameters for the metaG-analysis, such as what is the k-mer for assembling?

Information added in the text : Assembly was performed using metaSPAdes v.3.14.0 with kmers -k 21, 33, 55. Most of the metagenomic analysis parameters have been given in the announcements of the https://journals.asm.org/doi/full/10.1128/MRA.01253-20.

6. Line 215 and Line 217: SO4? SO4-?

The charges for the ions have been skipped to avoid confusion because we are talking about analysed total concentrations not different specifically charged ions. With ICP-OES or ICP-MS we measured total amount of elements, e.g. Sr or Cu, but not their specific ions. One can't differentiate between species by the above techniques.

7. Too many colors, plz show the top taxonomies or the most important taxonomies.

As instructed by reviewer 1, the taxonomy excel table has been made available online for easy viewing.

8. Sample names "MBS MB 1234" is confusing, plz use more readable ID.

Samples names MB were basically to distinct the biofilms samples collected around the copper plume from the other samples MBS. The denotations of the samples have been already used in the NCBI databases (as prerequisite for publication) and cannot be changed anymore.

9. Figure 7 is too complex, plz show the most important information.

Figure 7 is modified as instructed by first reviewer, the GO terms are simplified.

10. Fig8 and 9: The proposed pathway comes from all MAGs or just one MAG? What is the difference from already reported Copper-resistance pathway?

The proposed pathways are collected from all MAGs pathways map. Since the pathways are cumulative from the mixed microbiome, both copper-resistance pathways, the anaerobic Cus system and the aerobic CopA regulon are being observed. Moreover the sensors—CueR in *Salmonella*, and CsoR and RicR in *Mycobacterium*—which induce the expression of a number of copper-resistance mechanisms to counteract Cu toxicity and ensure survival are being observed.
https://www.researchgate.net/publication/283819561_Bacterial_Copper_Resistance_and_Virulence.

---

## Author Response (AR2)

Reviewer 1

    1.  L. 23: should the Hg have a charge?

The ionic forms are reduced to $Hg^0$. The ionic forms $Hg^{+2/+3}$ are not volatile, but the elemental Hg is volatile.

2. L. 78 and elsewhere: use correct notation for small subunit genes. The "S" is a unit of measure and always capitalized. Also, rRNA must be followed by "gene" when based on DNA.
Corrected

3. L. 103 and elsewhere: it is very hard to tell that the notation "MB1-6" is referring to 6 different samples. The way the MBS1-MBS4 samples are mentioned in line 94 is much easier to read. I'm the results it was very confusing that you were referring to multiple samples and hard to track the names to the figures.
MB1-6 changed to MB1-MB6 and the samples are grouped in () for convenience. The previous announcements and sequence submission follow the same nomenclature so at this point it is not advisable to rename/change it anymore.

4. L. 143: it really would be fine to leave the PCR details here.
Corrected

5. L. 212-224 and elsewhere: sulfate and nitrate are ions and must have their charges presented.
Corrected

6. L. 230-232: chao1 index is not shown in figure 4. Update figure 4 with an axis label for panel B and the caption for accuracy. Also correct the text so it matches the figure
Corrected

7. L. 235-248: I still find this section to be too strong. "Conclusively" and "replaced" are very definitive terms and I don't think you can be that strong without temporal data. To be this strong you need to demonstrate selection. Toning down the language and "suggesting " or "hypothesizing" conclusions will make your paper stronger while overstating your findings can be a red flag.
Corrected

8. Figure 4c: can any geochemical vectors be added? Also, consider using different shapes in your plots to make the figures colorblind accessible.
Actually, it is not possible because we don't have reading for all of the samples.

9. L. 268 and elsewhere: where are the DEseq2 data? These should be presented in the SI at least.
Added

10. L. 270-271: why are Deltaproteobacteria mentioned twice?
Typo corrected.

11. L. 271: correct "are"

corrected

12. L. 272: why are 2 values presented for Actinobacteria? Also in this paragraph please reference the figures.
These are values per sample MBS1-MBS4. Actinobacteria values in MBS3 and MBS4 samples. corrected

13. L. 275: add respectively after group
corrected
14. L. 276: assumed is too strong, hypothesized is more appropriate
corrected
15. L. 295-296: move reference to figure after observed.
corrected
16. L. 299: "accounts" is too strong. Your data suggests this but you're not sure because you didn't measure succession.
corrected
17. L. 301-312: why keep these data here? In the previous version I suggested edits to this section and to move it for flow. The responses stated that the changes were made but they weren't. If you disagree with the suggestion a response explaining why would be appreciated.
Your previous comment was 21. 302-313: the phylogenetic analysis was based on 350 bp amplicon sequences which doesn't provide a lot of information for robust taxonomic affiliation or phylogenetic inference. Making definitive statements about an OTU being rare or a novel class/species is a big reach based on limited sequence read length.
Yes, we agree that the phylogenetic analysis is based on 350 bp but it raises important questions about reclassifying the C0119 taxa as another class of Chloroflexi rather than in *Ktedonobacteria* and general comparison between the abundant taxas at species level is not possible anywhere else. The language has been toned down to avoid any novel class/species claims.
18. L. 323: specific
corrected
19. L. 345 and 346: wondering if these are the best word choices? Does a MAG really mediate a process or encode it? Or does the MAG have genes mediating or encoding for processes? (Also applied elsewhere)
corrected
20. Figure 7: please correct the blurry and overlapping text. Also GO or Go terms?
corrected
21. Reminder to make sure gene names are italicized.
corrected
22. L. 411-412: put the element symbols in parentheses
corrected

23. L. 425: "generally, the habitats" is unclear / hanging thought. What habitats? How are you distinguishing which habitats are dominated by chemolithoautotrophs?
Added "Acid mine drainage habitats"

24. L. 339-440: I think you should expand on which taxa are known to be metal resistant in your amplifying dataset.

Expanded

25. The paper is still missing comparisons to other mining affected systems. It would be valuable to know how often their key organisms are found in other heavy metal contaminated sites. Especially because Ktedonobacteria are poorly represented in culture. If this is a unique habitats it could be a great place to target for future cultivation work.

Ktedonobacteria are soil inhabiting bacteria which become abundant under heavy metal stress. L. 460-479 highlight this point. We tried to find *Ktedonobacteria* in similar mines settings but in vain; probably because of artificial illumination at the sampling site and cocolonization of *Ktedonobacteria* and *Oxyphotobacteria* makes this site a unique habitat. Furthermore, I had compared *Ktedonobacteria* MAG019 with other type strains (*K.racemifer* and *T.hazakensis*) in my thesis and found that MAG019 has some unique genes related to heat shock, copper homeostasis, etc Added in L.485

26. Did you omit Table 3? It's mentioned in the responses but no longer in the manuscript
No, the manuscript doesn't have any table, only supplementary info has tables and Table S3 is included in the URL https://doi.org/10.25625/DFFZ9R which may have caused confusion. The table S3 is changed to abundance graphs.

27. Figure S2: correct the spelling of proteobacteria and the overlapping text on the left
Corrected

28. Figure S3: pie charts are relative abundance?
Pie charts show the cumulative actual abundance of all taxes from all samples.
Corrected

29. Table S1 does not include the indices, only results of a statistical test. Please include both
Added

30. Table S2: define abbreviations especially for groups. Also recommend defining the different sub tables via a,b,c etc and including some borders
Corrected

31. PCR methods in SI are so short that it makes more sense to move them to the main pap
Corrected

Reviewer 2
The authors have addressed all the comments. I only have some technical concerns. Line 212,  223-228,  It may be better to add the ordinate axis and the tick marks in Figure 3.
Added

Line 214-221, The units "mgL-1" and "ugL-1" should be "mg·L-1" and "ug·L-1". Please check the full text and revise.
Corrected

Line 283-288, What kind of bacteria is "Delta Gamma Proteobacteria" in "MBS1" in Figure 5? Is there a typo? Please check and fix.
These are subclasses of Proteobacteria: Delta relates to the blue part of the column, Gamma to the green part so they were mentioned together in figure to show relative proportion.

Line 323, Some microorganism names in the article are not italicized, such as "Ktedonobacteria" on line 323. Please check and correct any typos in the text
Corrected